# Elastic Decision Transformer

**Yueh-Hua Wu**[12]    **Xiaolong Wang**[1*]    **Masashi Hamaya**[2*]
[1]UC San Diego    [2]OMRON SINIC X
yuw088@ucsd.edu

## Abstract

This paper introduces Elastic Decision Transformer (EDT), a significant advancement over the existing Decision Transformer (DT) and its variants. Although DT purports to generate an optimal trajectory, empirical evidence suggests it struggles with trajectory stitching, a process involving the generation of an optimal or near-optimal trajectory from the best parts of a set of sub-optimal trajectories. The proposed EDT differentiates itself by facilitating trajectory stitching during action inference at test time, achieved by adjusting the history length maintained in DT. Further, the EDT optimizes the trajectory by retaining a longer history when the previous trajectory is optimal and a shorter one when it is sub-optimal, enabling it to "stitch" with a more optimal trajectory. Extensive experimentation demonstrates EDT's ability to bridge the performance gap between DT-based and Q Learning-based approaches. In particular, the EDT outperforms Q Learning-based methods in a multi-task regime on the D4RL locomotion benchmark and Atari games. Videos are available at: https://kristery.github.io/edt/.

## 1 Introduction

Reinforcement Learning (RL) trains agents to interact with an environment and learn from rewards. It has demonstrated impressive results across diverse applications such as game playing [23, 44], robotics [41, 40, 54], and recommendation systems [12, 1]. A notable area of RL is Offline RL [31], which employs pre-collected data for agent training and proves more efficient when real-time interactions are costly or limited. Recently, the conditional policy approach has shown large potentials in Offline RL, where the agent learns a policy based on the observed state and a goal. This approach enhances performance and circumvents stability issues related to long-term credit assignment. Moreover, the successful Transformer architecture [49], widely used in applications like natural language processing [52, 13, 8] and computer vision [14, 34], has been adapted for RL as the Decision Transformer (DT) [11].

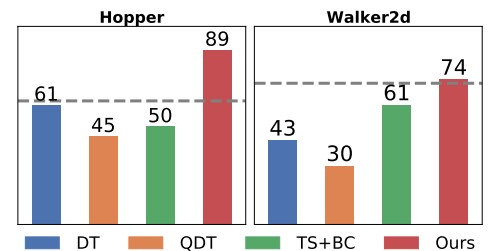

Figure 1: **Normalized return with medium-replay datasets.** The dotted gray lines indicate normalized return with **medium** datasets. By achieving trajectory stitching, our method benefits from worse trajectories and learns a better policy.

DT utilizes a Transformer architecture to model and reproduce sequences from demonstrations, integrating a goal-conditioned policy to convert Offline RL into a supervised learning task. Despite its competitive performance in Offline RL tasks, the DT falls short in achieving trajectory stitching, a desirable property in Offline RL that refers to creating an optimal trajectory by combining parts of sub-optimal trajectories [19, 9, 59]. This limitation stems from the DT's inability to generate

---

*equal advising

37th Conference on Neural Information Processing Systems (NeurIPS 2023).

superior sequences, thus curbing its potential to learn optimal policies from sub-optimal trajectories (Figure 1).

We introduce the Elastic Decision Transformer (EDT), which takes a variable length of the traversed trajectory as input. Stitching trajectories, or integrating the current path with a more advantageous future path, poses a challenge for sequence generation-based approaches in offline RL. Stitching a better trajectory appears to contradict one of the core objectives of sequence generation that a sequence generation model is required to reliably reproduce trajectories found within the training dataset. We suggest that in order to 'refresh' the prediction model, it should disregard 'negative' or 'unsuccessful' past experiences. This involves dismissing past failures and instead considering a shorter history for input. This allows the sequence generation model to select an action that yields a more favorable outcome. This strategy might initially seem contradictory to the general principle that decisions should be based on as much information as possible. However, our proposed approach aligns with this concept. With a shorter history, the prediction model tends to output with a higher variance, typically considered a weakness in prediction scenarios. Yet, this increased variance offers the sequence prediction model an opportunity to explore and identify improved trajectories. Conversely, when the current trajectory is already optimal, the model should consider the longest possible history for input to enhance stability and consistency. Consequently, a relationship emerges between the quality of the path taken and the length of history used for prediction. This correlation serves as the motivation behind our proposal to employ a variable length of historical data as input.

In practice, we train an approximate value maximizer using expectile regression to estimate the highest achievable value given a certain history length. We then search for the history length associated with the highest value and use it for action inference.

Evidence from our studies indicates that EDT's variable-length input sequence facilitates more effective decision-making and, in turn, superior sequence generation compared to DT and its variants. Furthermore, it is computationally efficient, adding minimal overhead during training. Notably, EDT surpasses state-of-the-art methods, as demonstrated in the D4RL benchmark [15] and Atari games [7, 17]. Our analysis also suggests that EDT can significantly enhance the performance of DT, establishing it as a promising avenue for future exploration.

**Our Contributions:**

- We introduce the Elastic Decision Transformer, a novel approach to Offline Reinforcement Learning that effectively addresses the challenge of trajectory stitching, a known limitation in Decision Transformer.

- By estimating the optimal history length based on changes in the maximal value function, the EDT enhances decision-making and sequence generation over traditional DT and other Offline RL algorithms.

- Our experimental evaluation highlights EDT's superior performance in a multi-task learning regime, positioning it as a promising approach for future Offline Reinforcement Learning research and applications.

## 2 Preliminaries

In this study, we consider a decision-making agent that operates within the framework of Markov Decision Processes (MDPs) [42]. At every time step $t$, the agent receives an observation of the world $o_t$, chooses an action $a_t$, and receives a scalar reward $r_t$. Our goal is to learn a single optimal policy distribution $P_\theta^*(a^t | o^{\leq t}, a^{<t}, r^{<t})$ with parameters $\theta$ that maximizes the agent's total future return $R_t = \sum_{k>t} r^k$ on all the environments we consider.

### 2.1 Offline Reinforcement Learning

Offline RL, also known as batch RL, is a type of RL where an agent learns to make decisions by analyzing a fixed dataset of previously collected experiences, rather than interacting with an environment in real-time. In other words, the agent learns from a batch of offline data rather than actively exploring and collecting new data online.

Offline RL has gained significant attention in recent years due to its potential to leverage large amounts of pre-existing data and to solve RL problems in scenarios where online exploration is

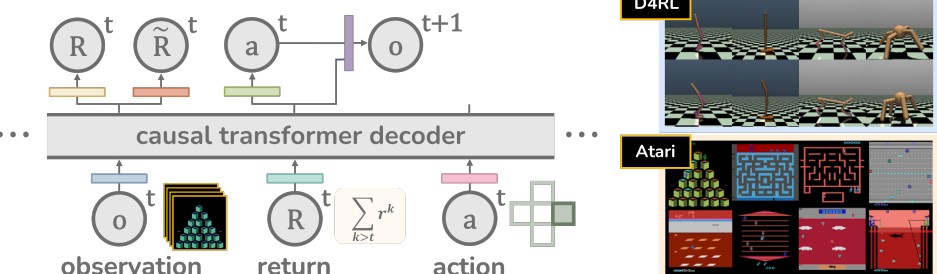

Figure 2: An overview of our Elastic Decision Transformer architecture. $\tilde{R}$ is the prediction of the maximum return. We also show the environments we used in the experiments on the right. We adopt four tasks for D4RL [15] and 20 tasks for Atari games.

impractical or costly. Examples of such scenarios include medical treatment optimization [33], finance [46], and recommendation systems [55].

Despite its potential benefits, offline RL faces several challenges, such as distributional shift, which occurs when the offline data distribution differs significantly from the online data distribution, and the risk of overfitting to the fixed dataset. A number of recent research efforts have addressed these challenges, including methods for importance weighting [35, 37], regularization [53, 28], and model-based learning [25, 29], among others.

## 2.2 Decision Transformer

The Decision Transformer architecture, introduced by [11], approaches the offline RL problem as a type of sequence modeling. Unlike many traditional RL methods that estimate value functions or compute policy gradients, DT predicts future actions based on a sequence of past states, actions, and rewards. The input to DT includes a sequence of past states, actions, and rewards, and the output is the next action to be taken. DT uses a Transformer architecture [49], which is composed of stacked self-attention layers with residual connections. The Transformer architecture has been shown to effectively process long input sequences and produce accurate outputs.

Despite the success of being applied to offline RL tasks, it has a limitation in its ability to perform "stitching." Stitching refers to the ability to combine parts of sub-optimal trajectories to produce an optimal trajectory. This approach can lead to a situation where the agent follows a sub-optimal trajectory that provides an immediate reward, even if a different trajectory leads to a higher cumulative reward over time. This limitation of DT is a significant challenge in many offline RL applications, and addressing it would greatly enhance the effectiveness of DT in solving real-world problems.

## 3 Elastic Decision Transformer

In this section, we present Elastic Decision Transformer (EDT), a model that automatically utilizes a shorter history to predict the next action when the traversed trajectory underperforms compared to those in the training dataset. The mechanism allows the model to switch to a better trajectory by forgetting 'unsuccessful' past experiences, thus opening up more possibilities for future trajectories. We further propose a method to estimate the maximum achieveable return using the truncated history, allowing EDT to determine the optimal history length and corresponding actions.

### 3.1 Reinforcement Learning as Sequence Modeling

In this paper, we adopt an approach to offline reinforcement learning that is based on a sequence modeling problem. Specifically, we model the probability of the next token in the sequence (denoted as $\tau$) based on all the tokens that come before it. The sequences we model can be represented as:

$$\tau = \langle ..., o^t, \hat{R}^t, a^t, ... \rangle,$$

where $t$ is a time step and $\hat{R}$ is the return for the remaining sequence. The sequence we consider here is similar to the one used in [30] whereas we do not include reward as part of the sequence and we predict an additional quantity $\tilde{R}$ that enables us to estimate an optimal input length, which we will cover in the following paragraphs. Figure 2 presents an overview of our model architecture. It should

be noted that we also change the way to predict future observation from standard DT [11], where the next observation is usually directly predicted from $a^t$ through the causal transformer decoder.

## 3.2 Motivation

We propose a shift in the traditional approach to trajectory stitching. Instead of focusing on training phases, we aim to achieve this stitching during the action inference stage. This concept is illustrated in Figure 3 using a simplified example. In this scenario, we consider a dataset, $D$, comprising only two trajectories: $D = (s_{t-1}^a, s_t, s_{t+1}^a), (s_{t-1}^b, s_t, s_{t+1}^b)$. A sequence model trained with this dataset is likely to predict the next states in a manner consistent with their original trajectories.

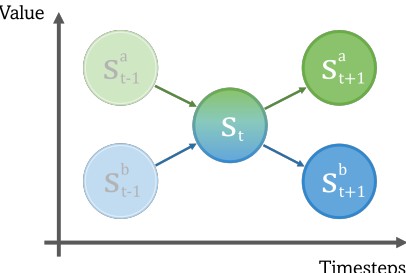

Figure 3: A Toy example to illustrate the motivation of EDT. The figure shows an offline RL dataset that contains only two trajectories $(s_{t-1}^a, s_t, s_{t+1}^a), (s_{t-1}^b, s_t, s_{t+1}^b)$.

To overcome this, we propose a method that enables trajectory stitching, where the model starts from $s_{t-1}^b$ and concludes at $s_{t+1}^a$. This is achieved by adaptively adjusting the history length. We introduce a maximal value estimator, $\tilde{R}$, which calculates the maximum value among all potential outcomes within the dataset. This allows us to determine the optimal history length that maximizes $\tilde{R}$.

In the given example, if the model starts at state $s_{t-1}^b$, it will choose to retain the history $(s_t)$ upon reaching state $s_t$, as $\tilde{R}(s_t) > \tilde{R}(s_{t-1}, s_t)$. Conversely, if the model initiates from state $s_{t-1}^a$, it will preserve the history $(s_{t-1}^a, s_t)$ when decision-making at $s_t$, as $\tilde{R}(s_{t-1}^a, s_t) \geq \tilde{R}(s_t)$. From the above example, we understand that the optimal history length depends on the quality of the current trajectory we've traversed, and it can be a specific length anywhere between a preset maximal length and a single unit.

To estimate the optimal history length in a general scenario, we propose solving the following optimization problem:

$$\arg\max_T \max_{\tau_T \in D} \hat{R}^t(\tau_T), \tag{1}$$

where $\tau_T$ denotes the history length $T$. More precisely, $\tau_T$ takes the form:

$$\tau_T = \langle o^{t-T+1}, \hat{R}^{t-T+1}, a^{t-T+1}, ..., o^{t-1}, \hat{R}^{t-1}, a^{t-1}, o^t, \hat{R}^t, a^t \rangle.$$

## 3.3 Training objective for Maximum In-Support Return

In the EDT, we adhere to the same training procedure as used in the DT. The key distinction lies in the training objective - we aim to estimate the maximum achievable return for a given history length in EDT. To approximate the maximum operator in $\max_{\tau_T \in D} \hat{R}^t(\tau_T)$, we employ expectile regression [38, 3], a technique often used in applied statistics and econometrics. This method has previously been incorporated into offline reinforcement learning; for instance, IQL [26] used expectile regression to estimate the Q-learning objective implicitly. Here, we leverage it to enhance our estimation of the maximum expected return for a trajectory, even within limited data contexts. The $\alpha \in (0, 1)$ expectile of a random variable $X$ is the solution to an asymmetric least squares problem, as follows:

$$\arg\min_{m_\alpha} \mathbb{E}_{x \in X} \left[ L_2^\alpha(x - m_\alpha) \right],$$

where $L_2^\alpha(u) = |\alpha - \mathbb{1}(u < 0)|u^2$.

Through expectile regression, we can approximate $\max_{\tau_T \in D} \hat{R}^t(\tau_T)$:

$$\tilde{R}_T^t = \max_{\tau_T \in D} \hat{R}^t(\tau_T) \approx \arg\min_{\tilde{R}^t(\tau_T)} \mathbb{E}_{\tau_T \in D}[L_2^\alpha(\tilde{R}^t(\tau_T) - \hat{R}^t)]. \tag{2}$$

We estimate $\tilde{R}^t$ by applying an empirical loss of Equation 2 with a sufficiently large $\alpha$ (we use $\alpha = 0.99$ in all experiments). The only difference in training EDT compared to other DT variants is the use of Equation 2, making the training time comparably shorter. We summarize our objective as:

$$\mathcal{L}_{EDT} = c_r \mathcal{L}_{return} + \mathcal{L}_{observation} + \mathcal{L}_{action} + \mathcal{L}_{max}, \tag{3}$$

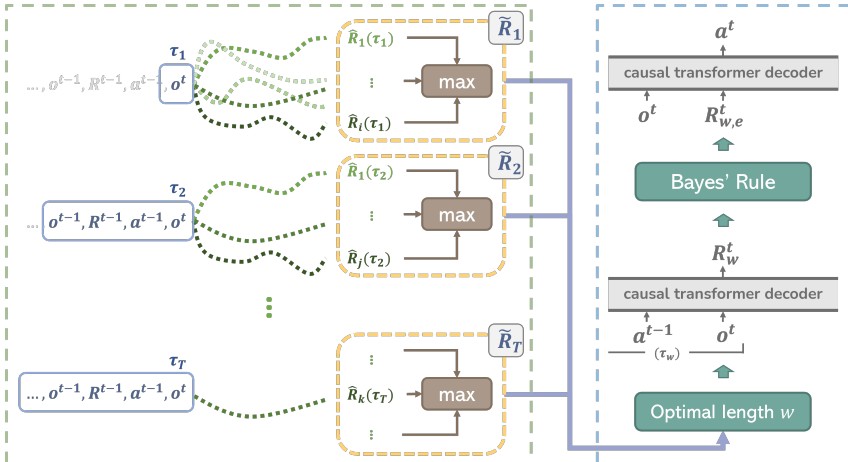

Figure 4: The figure illustrates the action inference procedure within the proposed Elastic Decision Transformer. Initially, we estimate the value maximizer, $\tilde{R}_i$, for each length $i$ within the search space, as delineated by the green rectangle. Subsequently, we identify the maximal value from all $\tilde{R}_i$, which provides the optimal history length $w$. Utilizing this optimal history length, we estimate the expert value at time step $t$, denoted as $\tilde{R}^t_{w,e}$, by Bayes' Rule. Finally, the action prediction is accomplished via the causal transformer decoder, which is indicated by the blue rectangle. In practice, we retain the distribution of $R^t_i$ during the estimation process for $\tilde{R}_i$ and we present the inference here for clarity.

---

**Algorithm 1** EDT optimal history length search

---

1: **Input:** a query sequence $\tau = \langle ..., o^{t-1}, R^{t-1}, a^{t-1}, o^t \rangle$ and EDT model $\theta_{EDT}$
2: **for** $w = 1, 1 + \delta, 1 + 2\delta, ..., T$ **do**
3:     Obtain $\tilde{R}^t(\tau^w)$ with truncated sequence $\tau^w = \langle o^{t-w+1}, R^{t-w+1}, a^{t-w+1}, ..., o^t \rangle$ and $\theta_{EDT}$
4: **end for**
5: Compute $P(R^t)$ with $\theta_{EDT}$ for $\tau^w$ that has the highest $\tilde{R}^t$ and then estimate $P(R^t|\text{expert}^t, ...)$ with Equation 4.

---

where $\mathcal{L}_{\text{observation}}$ and $\mathcal{L}_{\text{action}}$ are computed with a mean square error, $\mathcal{L}_{\text{return}}$ is a cross-entropy loss, and $\mathcal{L}_{\text{max}}$ is an empirical estimate of Equation 2. We set $c_r = 0.001$ to balance scale differences between mean square error and cross-entropy loss. In tasks with discrete action spaces like Atari, we optimize the action space as the tokenized return objective $\mathcal{L}_{\text{return}}$ using cross-entropy with weight $10c_r$. Our training method extends the work of [30] by estimating the maximum expected return value for a trajectory using Equation 2. This estimation aids in comparing expected returns of different trajectories over various history lengths. Our proposed method is not only easy to optimize, but can also be conveniently integrated with other DT variants. As such, it marks a significant advance in developing efficient offline reinforcement learning approaches for complex decision-making tasks.

### 3.4 Action Inference During Test time

During action inference phase in test time, we first (1) estimate the maximum achievable return $\tilde{R}_i$ for each history length $i$. Subsequently, (2) we predict the action by using the truncated traversed trajectory as input. The trajectory is truncated to the history length that corresponds to the highest value of $\tilde{R}_i$. These steps are elaborated in Figure 4.

To identify the history length $i$ that corresponds to the highest $\tilde{R}^t_i$, we employ a search strategy as detailed in Algorithm 1. As exhaustively searching through all possible lengths from 1 to $T$ may result in slow action inference, we introduce a step size $\delta$ to accelerate the process. This step size not only enhances inference speed by a factor of $\delta$, but also empirically improves the quality of the learned policy. An ablation study on the impact of the step size $\delta$ is provided in Appendix A. For all experiments, we set $\delta = 2$ to eliminate the need for parameter tuning.

To sample from expert return distribution $P(R^t, ...|\text{expert}^t)$, we adopt an approach similar to [30] by applying Bayes' rule $P(R^t, ...|\text{expert}^t) \propto P(\text{expert}^t|R^t, ...)P(R^t, ...)$ and approximate the distribu-

| Dataset | DT | QDT | TS+BC | S4RL | IQL | EDT (Ours) |
|---|---|---|---|---|---|---|
| hopper-medium | $60.7 \pm 4.5$ | $57.2 \pm 5.6$ | $\mathbf{64.3 \pm 4.2}$ | **78.9** | $63.8 \pm 9.1$ | $\mathbf{63.5 \pm 5.8}$ |
| hopper-medium-replay | $61.9 \pm 13.7$ | $45.8 \pm 35.5$ | $50.2 \pm 17.2$ | 35.4 | $\mathbf{92.1 \pm 10.4}$ | $\mathbf{89.0 \pm 8.3}$ |
| walker-medium | $71.9 \pm 3.9$ | $67.5 \pm 2.0$ | $78.8 \pm 1.2$ | **93.6** | $79.8 \pm 3.0$ | $72.8 \pm 6.2$ |
| walker-medium-replay | $43.3 \pm 14.3$ | $30.3 \pm 16.2$ | $61.5 \pm 5.6$ | 30.3 | $\mathbf{73.6 \pm 6.3}$ | $\mathbf{74.8 \pm 4.9}$ |
| halfcheetah-medium | $42.5 \pm 0.4$ | $42.3 \pm 2.5$ | $43.2 \pm 0.3$ | **48.8** | $47.3 \pm 0.2$ | $42.5 \pm 0.9$ |
| halfcheetah-medium-replay | $34.9 \pm 1.6$ | $30.0 \pm 11.1$ | $39.8 \pm 0.6$ | **51.4** | $44.1 \pm 1.1$ | $37.8 \pm 1.5$ |
| average | 52.5 | 45.5 | 56.3 | 56.4 | **66.7** | 63.4 |
| ant-medium | $92.5 \pm 5.1$ | - | - | - | $\mathbf{99.9 \pm 5.8}$ | $\mathbf{97.9 \pm 8.0}$ |
| ant-medium-replay | $\mathbf{87.9 \pm 4.9}$ | - | - | - | $91.2 \pm 7.2$ | $\mathbf{92.0 \pm 4.1}$ |
| average | 90.2 | - | - | - | **95.5** | **94.9** |

Table 1: Baseline comparisons on D4RL [15] tasks. Mean of 5 random training initialization seeds, 100 evaluations each. Our result is highlighted. The results of QDT, TS+BC, and S4RL are adopted from their reported scores. Following [26], we emphasize in bold scores within 5 percent of the maximum per task ($\geq 0.95 \cdot$ max).

tion of expert-level return with inverse temperature $\kappa$ [24, 48, 47, 43]:

$$P(R^t | \text{expert}^t, ...) \propto \exp(\kappa R^t) P(R^t). \tag{4}$$

While it may initially appear feasible to directly use the predicted $\tilde{R}$ as the expert return, it's important to note that this remains a conservative maximum operation. Empirically, we have found that Eq. 4 encourages the pursuit of higher returns, which consequently enhances the quality of the actions taken.

## 4  Experiments

Our experiments are designed to address several key questions, each corresponding to a specific section of our study:

- Does EDT significantly outperform DT and its variants? (Sec. 4.2, 4.3)
- Is the EDT effective in a multi-task learning regime, such as Locomotion and Atari games? (Sec. 4.3)
- Does a dynamic history length approach surpass a fixed length one? (Sec. 4.4)
- How does the expectile level $\alpha$ impact the model's performance? (Sec. 4.5)
- How does the quality of datasets affect the predicted history lengths? (Sec. 4.6)

We also provide an additional ablation study in Appendix A due to space constraints.

### 4.1  Baseline Methods

In the subsequent section, we draw comparisons with two methods based on the Decision Transformer: the original Decision Transformer (DT) [11] and the Q-learning Decision Transformer (QDT) [59]. Additionally, we include a behavior cloning-based method (TS+BC) [19], as well as two offline Q-learning methods, namely S4RL [45] and IQL [26], in our comparisons.

It is important to note that QDT and TS+BC are specifically designed to achieve trajectory stitching. QDT accomplishes this by substituting collected return values with estimates derived from Conservative Q-Learning [28]. Conversely, TS+BC employs a model-based data augmentation strategy to bring about the stitching of trajectories."

### 4.2  Single-Task Offline Reinforcement Learning

For locomotion tasks, we train offline RL models on D4RL's "medium" and "medium-replay" datasets. The "medium" dataset comes from a policy reaching about a third of expert performance. The ''medium-replay", sourced from this policy's replay buffer, poses a greater challenge for sequence modeling approaches such as DT.
We conclude our locomotion results in Table 1. Since the proposed model estimates the return of the current sequence, reward information is not required during test time.

Our observations indicate that the proposed EDT consistently outperforms the baseline DT and its variants on the majority of the datasets, with a notable performance advantage on the "medium-

| Task | DT-1 | IQL-1 | EDT-1 (Ours) |
|---|---|---|---|
| hopper | 51.2 | 59.8 | **76.9** |
| walker | 29.8 | 52.6 | **74.1** |
| halfcheetah | 30.5 | **40.4** | 36.8 |
| ant | 79.8 | 82.3 | **88.6** |
| sum | 191.3 | 235.1 | **276.4** |

Table 2: Evaluation results on multi-task regime. Mean of 5 random training initialization seeds, 100 evaluations each. The training dataset is a mixture of **medium-replay** datasets from the four locomotion tasks. Our main result is highlighted.

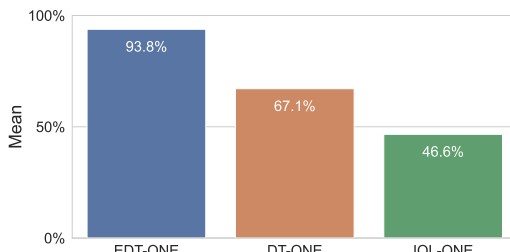

Figure 5: The average HNS comparison on 20 Atari games. The results are evaluated with three trials.

| Dataset | EDT (w=20) | EDT (w=5) | EDT (Ours) |
|---|---|---|---|
| hopper-m | **63.6 ± 5.2** | 57.8 ± 7.0 | **63.5 ± 5.8** |
| hopper-mr | 67.6 ± 27.7 | 62.6 ± 26.9 | **89.0 ± 8.3** |
| walker-m | 65.6 ± 11.7 | 62.6 ± 14.3 | **72.8 ± 6.2** |
| walker-mr | 64.5 ± 12.9 | 44.3 ± 8.7 | **74.8 ± 4.9** |
| halfcheetah-m | 42.0 ± 0.4 | 42.3 ± 0.8 | **42.5 ± 0.9** |
| halfcheetah-mr | 33.5 ± 8.0 | 36.4 ± 7.4 | **37.8 ± 1.5** |
| ant-m | 90.8 ± 16.0 | 95.6 ± 8.0 | **97.9 ± 8.0** |
| ant-mr | 80.0 ± 17.0 | 82.3 ± 15.9 | **92.0 ± 4.1** |
| sum | 507.6 | 483.9 | **570.3** |

Table 3: Mean of 5 random training initialization seeds, 100 evaluations each. In the Dataset column, "m" indicates medium, "mr" indicates medium-replay dataset. The "w" stands for history length. Our main results are highlighted.

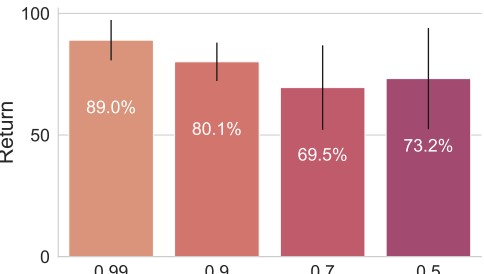

Figure 6: Ablation study on expectile level $\alpha$. Expectile objective reduces to Mean Square Error when $\alpha = 0.5$. Evaluated on Hopper and medium-replay dataset.

replay" datasets. These findings provide strong evidence that our approach is highly effective in stitching together sub-optimal trajectories with high return proportion, a task that DT and its variants cannot accomplish. Although EDT doesn't fully outperform IQL in the single-task learning, it does bridge the gap between Q-learning-based methods and DT by performing trajectory stitching with the estimated maximum return.

### 4.3 Multi-Task Offline Reinforcement Learning

This section aims to evaluate the multi-task learning ability of our model across diverse tasks, focusing on locomotion and Atari tasks. Locomotion tasks utilize vectorized observations, while Atari tasks depend on image observations. To emphasize the role of trajectory stitching, we restrict our datasets to medium-replay datasets for the four locomotion tasks and datasets derived from DQN Replay [2] for the Atari tasks. Our evaluations span 20 different Atari tasks, with further environment setup details available in the Appendix.

**Locomotion.** In the locomotion multi-task experiment, we maintain the same model architecture as in the single-task setting. By confining the dataset to **medium-replay** datasets from four tasks, we increase task complexity and necessitate the offline RL approach to learn and execute these tasks concurrently, while effectively utilizing trajectories generated by random policies. As depicted in Table 2, our proposed EDT successfully accomplishes all four tasks simultaneously without much performance compromise.

**Atari.** For Atari, we adopt a CNN image encoder used in DrQ-v2 [59] to process stacks of four 84x84 image observations. To ensure fair comparisons, all methods employ the same architecture for the image encoder. Following [30], we incorporate random cropping and rotation for image augmentation. Additional experiment details are delegated to the Appendix for brevity. Performance on each Atari game is measured by human normalized scores (HNS) [36], defined as (score − $\text{score}_{\text{random}}$)/($\text{score}_{\text{human}}$ − $\text{score}_{\text{random}}$), to ensure a consistent scale across each game.

Our experimental results align with those of [30], highlighting that Q-learning-based offline RL approaches encounter difficulties in learning a multi-task policy on Atari games. Despite IQL achieving the highest score in Table1, it demonstrates relative inadequacy in simultaneous multi-task learning as indicated in Table 2 and Figure 5. We leave the raw scores of the 20 Atari games in Appendix B.

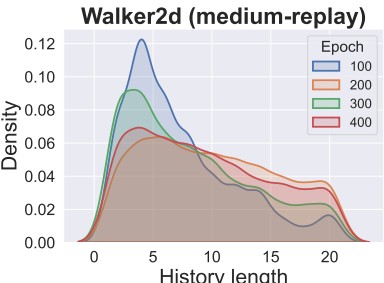
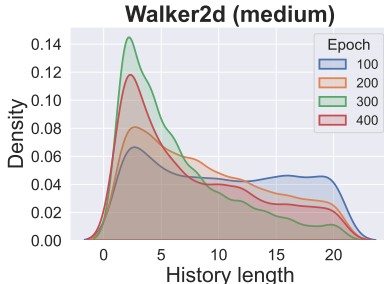

Figure 7: The figures illustrate the history length distributions across datasets and training epochs. For each distribution, we collect ten trajectories and derive a histogram of history lengths. The distribution is computed with kernel distribution estimate for better visualization.

## 4.4 Dynamic History Length vs. Fixed History Length

In Sec. 3.2, we proposed the concept of EDT, which adjusts history length based on the quality of the current trajectory. We illustrated this idea with a toy example.

To validate the benefits of this dynamic history length approach, we tested the EDT using both fixed and variable history lengths. The results, summarized in Table 3, show that the variable history length outperforms the fixed ones, particularly on the "medium-replay" datasets.

These findings suggest that the EDT effectively chooses a history length that yields a higher estimated return. While a shorter history length aids in trajectory stitching, it's also crucial to retain a longer history length to ensure the continuity of optimal trajectories. Therefore, the dynamic adjustment of history length in the EDT is key to its superior performance.

## 4.5 Ablation Study on Expectile Level $\alpha$

A key component of EDT is the approximation of the maximal value using expectile learning, as our method depends on accurately estimating these maximal values to choose the optimal history length. Consequently, examining the change in performance relative to the expectile level, $\alpha$, provides insight into the necessity of correct history length selection for performance enhancement.

The results, as displayed in Figure 6, suggest that when expectile regression is able to accurately approximate the maximizer, specifically at higher expectile levels, we observe both a higher average performance and lower standard deviation. This suggests that accurate selection of history length not only stabilizes performance but also enhances scores. Conversely, as the expectile level approaches 0.5, the expectile regression's objective shifts towards a mean square error, leading to an estimated value that is more of a mean value than a maximal one. This change makes it a less effective indicator for optimal history length. As a result, we can see a deterioration in EDT's score as the expectile level drops too low, and an increase in standard deviation, indicating inconsistency in the selection of an effective history length.

## 4.6 Analysis of Optimal History Length Distribution

In our analysis, we examine the history length distributions across various datasets, as depicted in Figure 7. Our findings reveal that the medium-replay dataset, which amalgamates trajectories from multiple policies, yields a distribution closely approximating a uniform distribution. Conversely, the medium dataset, acquired through a singular stochastic policy, exhibits a history length distribution characterized by an increased density at lower history lengths. This observation can be attributed to the prevalence of analogous trajectories within the medium dataset, leading to more frequent occurrences of trajectory stitching than the "medium-replay" dataset. However, it is important to acknowledge that the gains derived from this type of trajectory stitching remain limited, as the trajectories stem from identical distributions. Although performance improvement is observed, as presented in Table 1, it is significantly less pronounced in comparison to the medium-replay dataset.

Contrary to initial expectations, trajectory stitching does not occur as frequently within the medium-replay dataset as within the medium dataset. In fact, the distinct policies within the medium dataset contribute to the reduced instances of trajectory stitching, as their respective state distributions differ from one another. The diversity within the dataset results in a limited number of mutual $s_t$

instances illustrated in Figure 3. Nevertheless, the proposed EDT method derives substantial benefits from trajectory stitching in this context. The EDT effectively avoids being misled by sub-optimal trajectories within the dataset, demonstrating its capacity to make better decisions regarding history lengths and actions that optimize the current return.

## 5 Related Work

**Offline Reinforcement Learning.** Offline RL has been a promising topics for researchers since sampling from environments during training is usually costly and dangerous in real-world applications and offline reinforcement learning is able to learn a better policy without directly collecting state-action pairs. Several previous works have utilized constrained or regularized dynamic programming to mitigate deviations from the behavior policy [39, 51, 27, 16, 56].

Decision Transformer and its variants [11, 30, 59, 58, 57] have been a promising direction for solving offline RL from the perspective of sequence modeling. Trajectory Transformer (TT) [22] models distributions over trajectories using transformer architecture. The approach also incorporates beam search as a planning algorithm and demonstrates exceptional flexibility across various applications, such as long-horizon dynamics prediction, imitation learning, goal-conditioned reinforcement learning, and offline reinforcement learning.

Recently, there has been a growing interest in incorporating diffusion models into offline RL methods. This alternative approach to decision-making stems from the success of generative modeling, which offers the potential to address offline RL problems more effectively. For instance, [18] reinterprets Implicit Q-learning as an actor-critic method, using samples from a diffusion parameterized behavior policy to improve performance. Similarly, other diffusion-based methods [21, 32, 50, 5, 10, 4] utilize diffusion-based generative models to represent policies or model dynamics, achieving competitive or superior performance across various tasks.

**Trajectory Stitching.** A variety of methods have been proposed to tackle the trajectory stitching problem in offline RL. The Q-learning Decision Transformer (QDT) [59] stands out as it relabels the ground-truth return-to-go with estimated values, a technique expected to foster trajectory recombination. Taking a different approach, [19] utilizes a model-based data augmentation strategy, stitching together parts of historical demonstrations to create superior trajectories. Similarly, the Best Action Trajectory Stitching (BATS) [9] algorithm forms a tabular Markov Decision Process over logged data, adding new transitions using short planned trajectories. BATS not only aids in identifying advantageous trajectories but also provides theoretical bounds on the value function. These efforts highlight the breadth of strategies employed to improve offline RL through innovative trajectory stitching techniques.

## 6 Discussion

**Conclusion.** In this paper, we introduced the Elastic Decision Transformer, a significant enhancement to the Decision Transformer that addresses its limitations in offline reinforcement learning. EDT's innovation lies in its ability to determine the optimal history length, promoting trajectory stitching. We proposed a method for estimating this optimal history length by learning an approximate value optimizer through expectile regression.

Our experiments affirmed EDT's superior performance compared to DT and other leading offline RL algorithms, notably in multi-task scenarios. EDT's implementation is computationally efficient and straightforward to incorporate with other DT variants. It outshines existing methods on the D4RL benchmark and Atari games, underscoring its potential to propel offline RL forward.

In summary, EDT offers a promising solution for trajectory stitching, enabling the creation of better sequences from sub-optimal trajectories. This capability can considerably enhance DT variants, leading to improved performance across diverse applications. We are committed to **releasing our code.**

**Limitations.** A potential direction for future improvement involves enhancing the speed at which EDT estimates the optimal history. This could make the method suitable for real-time applications that have strict time constraints. While this adaptation is an exciting avenue for future research, it falls outside the primary scope of this paper.

## Acknowledgement

This project is supported by a grant from JST ACT-X, grant number JPMJAX22AC.

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

# A    Additional ablation study

## A.1    Step Size Ablation

In Algorithm 1, we introduced the step size parameter $\delta$, acting as a balance between search granularity and inference speed. The effects of varying $\delta$ are depicted in Figure 8. To compute the history length search space, we commence with the maximum history length. For instance, if the maximum history length is $T = 20$ and the step size $\delta = 8$, the history length search space becomes $20, 12, 4$.

Figure 8 shows that a narrowed search space leads to a decline in return performance and an increase in standard deviation, corroborating results from Table 3. EDT with a restricted history search space behaves more like EDT with a fixed history length. We found that the EDT works best when we chose a search step of $4$.

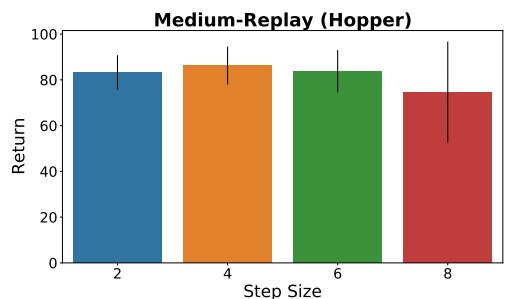

Figure 8: Ablation study on the step size $\delta$. The greater the step size the smaller the length search space is.

This could be because it is tough to make good estimates without being able to interact with the environment directly, as mentioned in previous studies [26, 28]. By increasing the step size, we were able to make our estimates more reliable and less affected by noise and other problems.

## A.2    History Length Distribution

We further show more history length distributions for the locomotion tasks here.

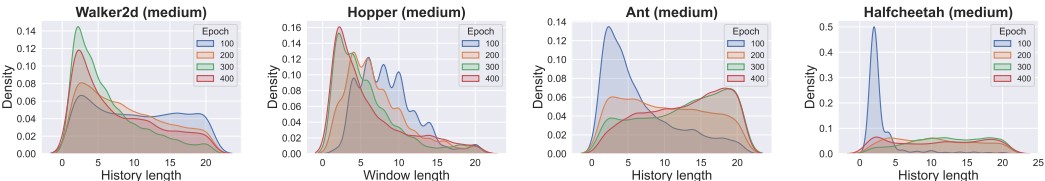

Figure 9: The figures illustrate the history length distributions across datasets and training epochs on the **medium** dataset. For each distribution, we collect ten trajectories and derive a histogram of history lengths. The distribution is computed with kernel distribution estimate for better visualization.

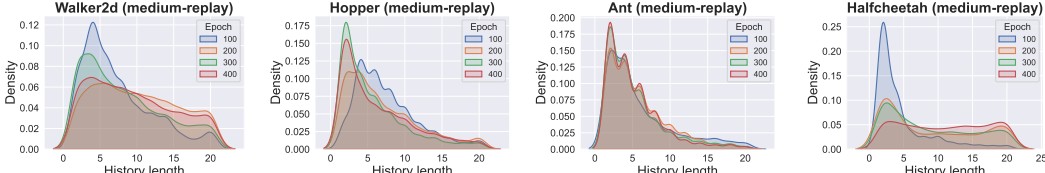

Figure 10: The figures illustrate the history length distributions across datasets and training epochs on the **medium-replay** dataset. For each distribution, we collect ten trajectories and derive a histogram of history lengths. The distribution is computed with kernel distribution estimate for better visualization.

Our initial hypothesis regarding the Ant task was that the medium-replay dataset would primarily consist of shorter history lengths, while the medium dataset would be more focused on longer history lengths. The distribution of history lengths in the medium and medium-replay datasets mostly supports this hypothesis.

In the medium-replay dataset, we consistently see a concentration of shorter history lengths. This was expected, given that the nature of the medium-replay dataset is likely to produce shorter history lengths. The distribution often converges towards higher densities for shorter lengths, which aligns with our expectations.

The pattern within the medium dataset, however, is less consistent. This could be attributed to several factors that can either elongate or truncate the history length. Despite these fluctuations, we still

observe a slight inclination towards longer history lengths. This meets our initial assumption but also demonstrates the complexity within the medium dataset.

### A.3 Comparison with Diffusion-Based Methods

# B Experiment Details

## B.1 Loss Computation Details

In this section, we detail the computation of the tokenized return loss $\mathcal{L}_{\text{return}}$ and the observation loss $\mathcal{L}_{\text{observation}}$ in Eq 1. For $\mathcal{L}_{\text{return}}$, we first tokenize the predicted return and the target return into $n$ bins bounded by the MAX and MIN scores obtained from D4RL [15]. We then compute the cross-entropy loss between the predicted value and the target value:

$$\mathcal{L}_{\text{return}} = CE(\hat{R}^t, R'^t), \tag{5}$$

where $R'^t$ suggests the predicted tokenized return. For $\mathcal{L}_{\text{observation}}$, we predict the next observation by minimizing the mean square error.

## B.2 Learning on Locomotion Tasks

In Section 4, we discussed the application of our approach to a multi-task learning scenario. Specifically, we consolidated medium-replay datasets from Ant, HalfCheetah, Hopper, and Walker2d. To standardize returns across these varied environments, we used scaling statistics (maximum return, minimum return, and return scale) from the official Decision Transformer repository (https://github.com/kzl/decision-transformer). As detailed in Section 3.3, we further segmented the return into 60 discrete buckets and, following the approach of [20, 30], sampled from the top 85th percentile logits within the discrete return distribution.

However, it's important to highlight that our return maximizer, $\tilde{R}$, estimates the scaled value directly rather than the discretized one. For the sequence $\langle ..., \mathbf{o}, R, a, ... \rangle$, we supplement each token embedding with a learned positional embedding. Given the differing state and action dimensions across tasks, we employ an additional state and action embedding layer. This layer transforms the raw state and action representation into a consistent state and action embedding size, while keeping the remainder of the architecture unchanged across tasks.

**Approximate $\tilde{R}$ Estimation.** Section 3.4 outlines how we use Bayes' Rule to estimate expert returns. A conventional $\tilde{R}$ prediction typically involves an autoregressive process, given the necessity of sampling from the discrete $\hat{R}_{\text{expert}}$ distribution at each time step during the search for the optimal history length. To simplify this process, we follow the approximation strategy used in [30]'s implementation. We mask all return values except the first one, thus making $\tilde{R}$ solely dependent on $\mathbf{o}$, $a$, and the initial return value.

**History Length Search Heuristic.** In Algorithm 1, we illustrated that a larger $\delta$ value allows us to infer actions more rapidly. Adopting this method needs to balance inference speed with search accuracy. Based on the concept introduced in Sec. 3.2, the optimal history length at the current state $s_{t+1}$ might be close to that of the previous state $s_t$. Therefore, given the optimal length $l_t$ at time step $t$, we search within the range $\{l_t - \Delta, l_t - \Delta + 1..., l_t + \Delta\}$ for the optimal length $l_{t+1}$ at the next step.

## B.3 Multi-Task Learning on Atari Games

The process for action inference in multi-task learning on Atari games closely aligns with that described in Sec. B.2, including the method for approximating $\tilde{R}$. However, there are several noteworthy distinctions, which we elaborate upon in this section.

Given that all games utilize grayscale frames of the same dimensions (84x84), we do not need to implement an additional state embedding layer as we did in the Locomotion scenario. Instead, we introduce a shared image encoder for all games, with further details outlined in Sec. B.5.

It's important to note the distinction between the action spaces of Atari games and Locomotion tasks. The action space of Atari games is discrete, so our sampling approach mirrors how we sample from the return distribution: we select from the top 85th percentile of action logits. Inspired by [30], we

discretize the reward signal into $\{-1, 0, +1\}$, while the return is split into 120 buckets ranging from $\{-20, ..., 100\}$.

Given the complexity and potential pitfalls of learning on 20 Atari games from scratch, we use a subset of GPT-2 to initialize the transformer decoder. This step improves both the convergence rate and the quality of the learned policy. Our dataset consists of 2 training runs from [2], with each run featuring rollouts from 50 checkpoints. Lastly, we enhance the dataset with image augmentations, including random cropping and rotation.

## B.4 Atari Games

Due to time and computational constraints, we randomly selected 20 tasks from the 41 tasks in the study by [30]. Details about the game types and number of action spaces can be found in Table 4. We also provide the raw scores of the Atari experiments in Table 5. Given that we transform the original observations to grayscale and rescale them to 84x84 images, we illustrate these transformed observations in Figure 11.

Table 4: Atari Games: Name, Category, and Action Space

| Game | Category | Action Space |
|---|---|---|
| Assault | Shooter | 7 |
| Asterix | Platform | 18 |
| BankHeist | Strategy | 18 |
| BeamRider | Shooter | 9 |
| Breakout | Arcade | 4 |
| Centipede | Shooter | 18 |
| ChopperCommand | Shooter | 18 |
| Enduro | Racing | 9 |
| FishingDerby | Sports | 18 |
| Freeway | Racing | 3 |
| Frostbite | Platform | 18 |
| Gravitar | Shooter | 18 |
| NameThisGame | Shooter | 8 |
| Phoenix | Shooter | 8 |
| Qbert | Puzzle | 6 |
| Seaquest | Shooter | 18 |
| TimePilot | Shooter | 10 |
| VideoPinball | Pinball | 3 |
| WizardofWor | Shooter | 10 |
| Zaxxon | Shooter | 18 |

Table 5: Raw Atari scores for the three offline RL approaches. The scores are average results over three trials. The Human Avg and Random scores are adopted from [6].

| Games | Human Avg | Random | EDT-ONE (80M) | DT-ONE (80M) | IQL-IQM (80M) |
|---|---|---|---|---|---|
| Asterix | 8503.3 | 210 | 12089.6 | 7731.0 | 994.6 |
| Assault | 742.0 | 222.4 | 1849.2 | 1260.0 | 964.3 |
| BankHeist | 753.1 | 14.2 | 5.0 | 9.3 | 21.3 |
| BeamRider | 16926.5 | 363.9 | 6469.9 | 4864.6 | 3865.3 |
| Breakout | 30.5 | 1.7 | 220.1 | 180.8 | 70.9 |
| Centipede | 12017.0 | 2090.9 | 2164.0 | 2290.4 | 2940.4 |
| ChopperCommand | 7387.8 | 811 | 3491.6 | 2774.8 | 331.7 |
| Enduro | 860.5 | 0 | 823.6 | 557.1 | 877.8 |
| FishingDerby | -38.7 | -91.7 | -1.7 | -43.3 | 30.6 |
| Freeway | 29.6 | 0 | 26.0 | 13.5 | 37.8 |
| Frostbite | 4334.7 | 65.2 | 1837.6 | 1558.1 | 488.5 |
| Gravitar | 3351.4 | 173 | 70.5 | 119.8 | -37.1 |
| NameThisGame | 8049.0 | 2292.3 | 7529.4 | 5939.7 | 3954.3 |
| Phoenix | 7242.0 | 761.4 | 4280.5 | 3369.6 | 1893.1 |
| Qbert | 13455.0 | 163.9 | 11739.8 | 7978.4 | 11035.5 |
| Seaquest | 42054.0 | 68.4 | 4410.9 | 2628.3 | 756.7 |
| TimePilot | 5229.2 | 3568.0 | 2737.7 | 3036.7 | 2311.3 |
| VideoPinball | 17667.9 | 0 | 875.7 | 476.0 | 0 |
| WizardOfWor | 4756.5 | 563.5 | 222.2 | 384.8 | 498.2 |
| Zaxxon | 9173.3 | 32.5 | 233.4 | 141.7 | -6.5 |

## B.5 Image Encoder

To ensure a fair comparison, we standardize the image encoder across EDT and the two baseline approaches. We adopt the image encoder from DrQ-v2 [60] for this purpose.

The architecture of the **image encoder** is as followed:

- 1 convolution with stride 2, output channels 32, kernel size 3. (ReLU).

- 3 convolution with stride 1, output channels 32, kernel size 3. (ReLU).
- 1 fully connected layer and $H$ output dimensions.

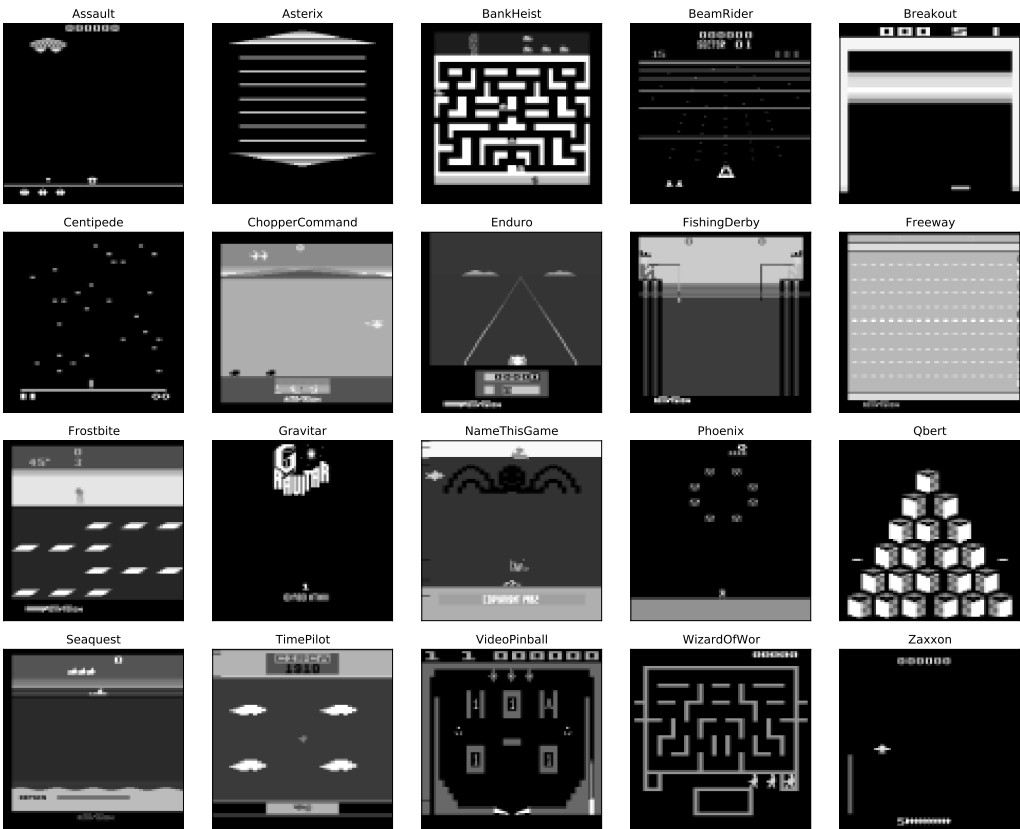

Figure 11: All Atari games used for our experiments. We adopted gray scale and reshape observations to images of size 84x84.

## C  Inter-Quartile Mean of the Atari results

Alongside the main paper's results, we present the Inter-Quartile Mean (IQM) of our findings. Notably, IQL outperforms DT in terms of IQM. This occurs due to the variance in task difficulties, despite our use of HNS to balance scales across tasks. The difficulty disparity among games leads to significant variance. However, despite this, our proposed EDT still significantly surpasses both baselines, demonstrating its robust performance.

In the original Decision Transformer model, estimating the remaining return-to-go is dependent on a preset expected return-to-go value. This requirement presents a challenge when applied to multi-task environments, where return-to-go values can significantly vary across different tasks.

The issue is particularly pronounced in Atari games where high scores often necessitate long trajectories. These long trajectories may consequently yield negative

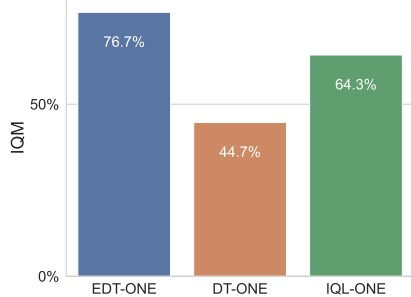

Figure 12: Results of learning from 20 Atari games in terms of IQM.

estimates for the return-to-go, which presents an issue for the model's performance and application.

Thus, the unique characteristics of different tasks highlight the limitations of a one-size-fits-all approach in the context of return-to-go estimation in the Decision Transformer model.

## D Architecture Details

We adopt the causal transformer decoder architecture and process the sequences as follows.

- **Transformer Blocks:** Composed of masked causal attention and a multilayer perceptron (MLP), with layer normalization and residual connections. Activation function used is GELU (Gaussian Error Linear Unit).
- **Embedding Layers:**
    - State Embedding: A linear layer followed by positional (time) embeddings addition.
    - Action Embedding: A linear layer followed by positional (time) embeddings addition.
    - Return to Go Embedding: A linear layer followed by positional (time) embeddings addition.
    - Timestep Embedding: An embedding layer for encoding timesteps.
- **Prediction Heads:**
    - State Prediction: A linear layer taking as input the concatenated action and state embeddings.
    - Action Prediction: A linear layer followed by a Tanh activation function.
    - Return-to-go Prediction: A linear layer.

Table 6: Hper-parameters. We list all hyper-parameters for D4RL and Atari games below.

| Parameter | Setting (D4RL) | Setting (Atari) |
|---|---|---|
| Observation down-sampling | - | 84 x 84 |
| Frames stacked | - | 4 |
| Frames skip | - | 4 |
| Terminal on loss of life | - | True |
| Augmentation (random cropping) | - | True |
| Augmentation (random rotation) | - | True |
| Discount factor | 1 | $0.997^4$ |
| Gradient clipping | True | True |
| Reward clipping | True | True |
| Optimizer | AdamW | AdamW |
| Optimizer (learning rate) | 0.0001 | 0.0001 |
| Optimizer (weight decay) | 0.0001 | 0.0001 |
| Maximum history length | 20 | 30 |
| inverse temperature ($\kappa$) | 10 | 10 |
| Expectile level | 0.99 | 0.99 |
| Mix precision | True | True |
| Evaluation episodes | 100 | 32 |
| Action loss coefficient ($\mathcal{L}_{\text{action}}$) | 1 | 0.01 |
| Return loss coefficient ($\mathcal{L}_{\text{return}}$) | 0.001 | 0.001 |
| State loss coefficient ($\mathcal{L}_{\text{state}}$) | 1 | 1 |
| Return maximizer loss coefficient ($\mathcal{L}_{\text{max}}$) | 0.5 | 0.5 |
| Number of return output bins | 60 | 120 |
| Observation normalization | True | True |
| Batch size | 256 | 256 |
| Sample from the top 85th percentile logits (return) | True | True |
| Sample from the top 85th percentile logits (action) | - | True |
| Step size ($\delta$) | 2 | 2 |
| GPU | 3090 Ti | V100 |

