# OpenReview forum: "Elastic Decision Transformer"
_NeurIPS.cc/2023/Conference — NeurIPS 2023 poster_

### Official Review · Reviewer_nm2U · 2023-07-02

**Soundness:** 3 good
**Presentation:** 4 excellent
**Contribution:** 2 fair
**Rating:** 6
**Confidence:** 4

**Summary:**

This paper tackles the "trajectory stitching" problem of Decision Transformer. It proposes a method called Elastic Decision Transformer (EDT) that dynamically adjusts the history at inference time such that longer history is maintained for optimal trajectories and shorter history is maintained for sub-optimal ones. Empirical evidence shows that EDT can better achieve trajectory stitching than original DT. Furthermore, it is able to outperform methods based on Q-Learning on certain D4RL tasks and Atari games.

**Strengths:**

- This paper is well presented and easy to follow.

- The idea to dynamically adjust history to achieve trajectory stitching is novel and well motivated.

- Experiment results show that the proposed method seems robust enough to benefit from optimal and sub-optimal trajectories.

- Code is submitted. Though I didn't run them.

**Weaknesses:**

- The applicability of the proposed method seems to be limited since it is only applicable to DT and its variants.

- Chosen baselines might not cover all recent progresses in offline RL, especially those specifically designed to address trajectory stitching.

- Comparison to recent methods based on Diffusion model, including Diffuser[Janner et al., 2022] and Decision Diffuser[Ajay et al., 2023], is missed.

- Though the paper claims EDT to be generally applicable for all DT variants, it is only evaluated against the original DT.

- Typo, L234 should be Table 2 instead of Figure 2.

References:
- Janner et al., Planning with Diffusion for Flexible Behavior Synthesis, ICML 2022.
- Ajay et al., Is Conditional Generative Modeling all you need for Decision Making?, ICLR 2023.

**Questions:**

- How is the method compared to diffusion-based planning methods for offline RL, such as Diffuser[Janner et al., 2022] and Decision Diffuser[Ajay et al., 2023]?

- Can we demonstrate how the EDT idea could improve the performance of other DT variants?

References:
- Janner et al., Planning with Diffusion for Flexible Behavior Synthesis, ICML 2022.
- Ajay et al., Is Conditional Generative Modeling all you need for Decision Making?, ICLR 2023.

**Limitations:**

See above.

---

> ### Author Rebuttal · Authors · 2023-08-08
>
> We thank the reviewer for their insightful comments. We address your comments in the following.
>
> ---
>
> **Q**: "The applicability of the proposed method seems to be limited since it is only applicable to DT and its variants."
>
> **A**: In this work, we primarily focus on improving DT and its variants. However, the motivation and intuition behind our approach are not confined solely to DT but extend to sequence generation models as well. For models that predict future actions or statistics and struggle with trajectory stitching, the intuition proposed in this work—namely, using a shorter history length for more accurate prediction results—could potentially be applicable.
>
> ---
>
> **Q**: “How is the method compared to diffusion-based planning methods for offline RL, such as Diffuser[Janner et al., 2022] and Decision Diffuser[Ajay et al., 2023]?”
>
> **A**:
> We present the performance comparison on D4RL in the Table below, including the reported values from Decision Diffuser [1] to ensure a fair comparison. While diffusion-based models sometimes show comparable or even superior performance in single-task scenarios, it's important to note that our EDT model demonstrates scalability across multi-game scenarios with a large amount of mixed demonstrations—an area where diffusion models are unexplored. Even a DT-based method like Multi-Game Decision Transformers [2], which does not directly address the trajectory stitching issue, substantially outperforms Q-learning-based approaches such as CQL.
>
> In our multi-task learning experiments, EDT maintains a clear advantage, achieving both superior scalability and performance. We believe these results highlight the unique contributions and potential applications of EDT.
>
> |                           | Diffuser [3] | Decision Diffuser [1] |   DT   |   EDT  |
> |:-------------------------:|:--------:|:-----------------:|:------:|:------:|
> |    hopper-medium-replay   |  $96.8$  |      $100.0$      | $61.9$ | $89.0$ |
> |    walker-medium-replay   |  $61.2$  |       $75.0$      | $43.3$ | $74.8$ |
> | halfcheetah-medium-replay |  $42.2$  |       $39.3$      | $34.9$ | $37.8$ |
> |       hopper-medium       |  $58.5$  |       $79.3$      | $60.7$ | $63.5$ |
> |       walker-medium       |  $79.7$  |       $82.5$      | $71.9$ | $72.8$ |
> | halfcheetah-medium-replay |  $44.2$  |       $49.1$      | $42.5$ | $42.5$ |
>
> **Q**: "Can we demonstrate how the EDT idea could improve the performance of other DT variants?"
>
> **A**: An example that illustrates the EDT idea in comparison to other DT variants can be seen in its application to the Online Decision Transformer (ODT) [4]. In ODT, action prediction is modeled as a stochastic policy, optimized by minimizing the negative log-likelihood (NLL) loss. To demonstrate the impact of EDT, we compare two methods: 1) Offline RL using stochastic action output, optimized with NLL loss, and 2) Offline RL incorporating EDT, but using the same stochastic action and NLL loss. Both methods follow standard DT training protocols. The experimental results, summarized in the table below, indicate that incorporating EDT into the architecture and training loss proposed by ODT leads to enhanced performance.
>
> |                        | hopper-medium | hopper-medium-replay |
> |:----------------------:|:-------------:|:--------------------:|
> |    ODT architecture    | $58.9\pm 5.1$ |    $65.3\pm 13.2$    |
> | ODT architecture + EDT | $62.3\pm 3.8$ |     $82.8\pm 8.1$    |
>
> ---
>
> Please do not hesitate to let us know if you have any additional comments.
>
> [1] Ajay et al., Is Conditional Generative Modeling all you need for Decision-Making? (2023), https://arxiv.org/abs/2211.15657
>
> [2] Lee et al., Multi-Game Decision Transformers (2022), https://arxiv.org/abs/2205.15241
>
> [3] Janner et al., Planning with Diffusion for Flexible Behavior Synthesis (2022), https://arxiv.org/abs/2205.09991
>
> [4] Zheng et al., Online Decision Transformer (2022), https://proceedings.mlr.press/v162/zheng22c.html

---

> > ### Comment · Reviewer_nm2U · 2023-08-15
> >
> > Thanks authors for addressing my questions. I would like to keep my score.

---

### Official Review · Reviewer_C3CW · 2023-07-04

**Soundness:** 3 good
**Presentation:** 3 good
**Contribution:** 3 good
**Rating:** 5
**Confidence:** 4

**Summary:**

This paper proposes the Elastic Decision Transformer (EDT), an improvement over the existing Decision Transformer (DT). DT has had notable limitations with trajectory stitching, i.e., generating optimal or near-optimal trajectories from sub-optimal ones. EDT addresses this issue by modifying the history length utilized in DT, optimizing the trajectory by maintaining a longer history for optimal trajectories and a shorter one for sub-optimal ones. This adjustment allows the model to "stitch" more optimal trajectories.

**Strengths:**

** Strengths**

1. The concept of Elastic Decision Transformer (EDT) offers a novel approach to tackle the issue of trajectory stitching.
2. The paper successfully translates the abstract theoretical concepts into practical, tested methods. The methodology of using varying history lengths based on trajectory quality is both innovative and intuitive.
3. The empirical results demonstrating EDT's superior performance in various benchmarks provide robust support for the proposed model.

**Weaknesses:**

** Weaknesses **
1. While the paper presents compelling evidence of EDT's superiority over the original DT, it would further enrich the discussion to include comparative analyses with other state-of-the-art DT variants. Examples of such variants include Prompt DT [1], Hyper DT [2], and Multi-Game DT [3].

2. A potential concern arises from the fact that the EDT's performance hinges heavily on the value maximizer. While effective in function, it tends to be less efficient in terms of training speed. Exploring and discussing ways to enhance this aspect could be beneficial.

3. The paper could benefit from a more elaborate explanation of the value maximizer's mechanics, particularly the implementation of expectile regression. Understanding this component better could aid readers in appreciating the model's inner workings.

[1] Xu, Mengdi et al. “Prompting Decision Transformer for Few-Shot Policy Generalization.” ArXiv abs/2206.13499 (2022): n. pag.
[2] Xu, Mengdi et al. “Hyper-Decision Transformer for Efficient Online Policy Adaptation.” ArXiv abs/2304.08487 (2023): n. pag.
[3] Lee, Kuang-Huei et al. “Multi-Game Decision Transformers.” ArXiv abs/2205.15241 (2022): n. pag.

**Questions:**

1. I recommend the authors include further results for comparison with the state-of-the-art algorithms I've referenced in the weaknesses section, namely Prompt DT, Hyper DT, and Multi-Game DT. Such comparative analysis would be instrumental in providing a clearer understanding of the proposed method's efficacy.

2. Upon careful scrutiny of the experimental results, I noted a lack of evaluations concerning the training speed associated with different step sizes denoted by "$\delta$". It would be worthwhile to explore and discuss how this parameter influences both the learning time and the overall performance of the model.

3. How does the EDT handle scenarios where the distinction between optimal and sub-optimal trajectories is not clear?

4. How would EDT perform with extremely long trajectories where the computation cost may become prohibitive?

5. Could you expand on how the computation overhead during training is minimized?

**Limitations:**

The major limitation of this paper is value maximizer training speed, as illustrated in the paper.
I don't find any obvious broader societal impacts.

---

> ### Author Rebuttal · Authors · 2023-08-08
>
> We thank the reviewer for their insightful comments. We address your comments in the following.
>
> ---
>
> **Q**: A potential concern arises from the fact that the EDT's performance hinges heavily on the value maximizer. While effective in function, it tends to be less efficient in terms of training speed.
> **A**: Although EDT employs an additional expectile regression loss compared to DT, the training time is not significantly affected. This is because the expectile regression loss involves only an additional one-layer MLP, and the computation is similar to that of MSE. Below, we compare the wall clock time for training both DT and EDT to provide a more comprehensive result. (3090 GPU, 64 batch size)
>
> |                                            |   DT   |   EDT  |
> |:------------------------------------------:|:------:|:------:|
> | wall clock training time (500 epochs, sec) | $1442$ | $1449$ |
>
>
> ---
>
> **Q**: “The paper could benefit from a more elaborate explanation of the value maximizer's mechanics, particularly the implementation of expectile regression.”
>
> **A**: Thank you for the constructive advice. During EDT training, the $\tilde{R}$ is trained with respect to different history lengths and this can be achieved with a standard data loader and DT training. To accomplish this, we sample sub-trajectories with a context length $T$. For example, let’s say $T=3$, and the sampled input will be $[o_{t-2}, R_{t-2}, a_{t-2}, o_{t-}, R_{t-1}, a_{t-1}, o_{t}, R_{t}, a_{t}]$, with the task being to predict $[(R_{t-2}, \tilde R_{t-2}), a_{t-2}, o_{t-1}, (R_{t-1}, \tilde R_{t-1}), a_{t-1}, o_{t}, (R_{t}, \tilde R_{t}), a_{t}, o_{t+1}]$. In this case, the model is trained to predict $R_{t-2}$ and $\tilde R_{t-2}$ using only $o_{t-2}$, which corresponds to a unit history length. Due to the inherent structure of DT training, EDT naturally learns the value of $\tilde{R}$ for each potential length. The implementation of expectile regression for $\tilde R$ is fundamentally the same as the implementation of MSE loss for $R$, with the exception that we use expectile regression (as defined in Eq. 2). This can be computed with just a few lines of Python code, as shown below:
> ```
> def expectile_loss(diff, expectile=0.99):
>    weight = torch.where(diff > 0, expectile, (1 - expectile))
>    return weight * (diff**2)
> ```
> We will add more details regarding the implementation of expectile regression in our revised manuscript.
>
> ---
>
> **Q**: “I recommend the authors include further results for comparison with the state-of-the-art algorithms I've referenced in the weaknesses section, namely Prompt DT, Hyper DT, and Multi-Game DT. Such comparative analysis would be instrumental in providing a clearer understanding of the proposed method's efficacy.”
>
> **A**: Thank you for your suggestions! The studies on Prompt DT and Hyper DT explore potential directions to enable a DT model to adapt to novel tasks. On the other hand, Multi-Game DT, which is similar to EDT, is focused on large-scale multi-task learning with DT. We will include Prompt DT and Hyper DT in our related work section to provide a more comprehensive survey.
>
> ---
>
> **Q**: “It would be worthwhile to explore and discuss how parameter $\delta$ influences both the learning time and the overall performance of the model.”
>
> **A**: Thank you for pointing this out. The parameter $\delta$ does not influence the learning time as action inference with varying history length is not required during training. This, however, does not imply that the training length is fixed. Instead, the value $\tilde{R}$ is learned for each history length. We incorporate the parameter $\delta$ only in the evaluation phase, where the history length search is performed to find the optimal length. In the Table below, we summarize the ablation study on the value of $\delta$ and the corresponding inference time on a 3090 GPU.
>
> |               $\delta$               |    2   |    4   |    6   |    8   |   DT   |
> |:------------------------------------:|:------:|:------:|:------:|:------:|:------:|
> | wall clock time for 1000 steps (sec) | $12.6$ | $12.1$ | $11.8$ | $11.5$ | $11.4$ |
>
> ---
>
> **Q**: How does the EDT handle scenarios where the distinction between optimal and sub-optimal trajectories is not clear?
>
> **A**: It is actually easier for EDT to combine optimal parts of trajectories when the distinction between optimal and sub-optimal trajectories is not clear. This is because, in such a case, EDT can identify more common or similar parts between these trajectories and then switch to the one with a higher estimated return.
>
> ---
>
> **Q**: How would EDT perform with extremely long trajectories where the computation cost may become prohibitive?
>
> **A**: In our implementation, instead of keeping all the previous history, we maintain a maximum history length $T$ to store previous trajectories. In our experiments, we use $T=20$ for D4RL tasks and $T=30$ for Atari experiments. We can alleviate the computation cost by either using a shorter $T$ or a greater $\delta$ as proposed in the paper, which will reduce the number of queries for each action inference.
>
> ---
>
> **Q**: Could you explain how the computation overhead during training is minimized?
>
> **A**: The additional computational overhead in training EDT stems solely from the expectile regression used for maximum return estimation. This loss calculation is efficiently carried out through the expectile regression loss (Eq. 2) using an additional one-layer MLP, as illustrated in Figure 2. As the EDT’s training only involves this additional expectile regression loss, its training time is almost identical to that of the vanilla DT. Therefore, the complexity does not significantly impact the efficiency of our method, enabling us to minimize the computational overhead.
>
> ---
>
> Please do not hesitate to let us know if you have any additional comments.

---

> > ### Comment · Reviewer_C3CW · 2023-08-17
> >
> > I thank the authors for the detailed answers.  I'd like to keep the score.

---

> ### Author Response · Authors · 2023-08-15
>
> Dear Reviewer C3CW,
>
> Thank you once again for your feedback and comments on our work! We would appreciate it if you could let us know if there are any further questions or concerns.
>
> Best regards,
>
> Author 134

---

### Official Review · Reviewer_ak2g · 2023-07-07

**Soundness:** 4 excellent
**Presentation:** 3 good
**Contribution:** 4 excellent
**Rating:** 7
**Confidence:** 4

**Summary:**

This paper proposes Elastic Decision Transformer (EDT), which helps Decision Transformer (DT) generalize offline data by trajectory stitching. Their changes to the DT are: 1) during training, adding an extra term to the objective to estimate the maximum Q values starting from a state (similar to the Implicit Q-Larning algorithm in the literature); 2) during inference, it determines the optimal length of trajectory to put in the prompt to most-likely generate a trajectory with a high return. EDT is evaluated with other state-of-the-art offline RL algorithms (including both classical and DT-based ones) on D4RL and Atari domains.

**Strengths:**

This work addresses a well-known problem in DT, the ability to generalize offline data by stitching different trajectories. There are earlier works that try to address the same problem (Q-Learning Decition Transformer, for example), but the earlier works do not achieve as good performance as EDT.

This work brings the performance of DT-based offline RL methods to match the best algorithms in the literature (IQL). The authors did a thorough evaluation of EDT and baseline algorithms, on D4RL and Atari datasets, which are common benchmarks for offline RL.

The modifications to DT are simple, well-justified, and effective. The paper is overall clearly written and easy to follow.


**Weaknesses:**

**Contributions.** Does this simply implement the idea behind the IQL algorithm in DT? If this is the case, it should be emphasized somewhere in the paper. I have some questions about this in the question section below.

**Clarity.** Need background on expectile regression. It may be helpful to include a figure like Fig. 1 (left) in the IQL paper (Kostrikov et al., 2021) to make this paper more self-competent.

**Minor points.**

Line 94: DT uses rewards-to-go, or return, not rewards.

Line 205: Extra quote mark at the end of the line.

Structure of Sec. 3: Line 115: \tilde{R} should be clearly defined here, it’s a “maximal value estimator.” It’s defined on a trajectory segment.

It would be great to change the color of Fig. 3. The node labels ($S^a_{t-1}$ and $S^b_{t-1}$) are a bit hard to read.

**Missing references**

The paper considered the multi-task regime, but some related works are missing:

Reed, Scott, et al. "A generalist agent." arXiv preprint arXiv:2205.06175 (2022).

Xu, Mengdi, et al. "Prompting decision transformer for few-shot policy generalization." international conference on machine learning, 2022.


**Questions:**

My understanding is that training always uses the same history length ($T$). Does it help to use different history lengths during training as well? It seems that training and inference use histories of different lengths (always length of $T$ for training, length of $\leq T$ for inference). Would it harm the performance?

Sec. 4.4 shows that different history lengths do affect the performance. I wonder if the optimal history length search can be avoided without harming the performance: Given the estimate of the maximum value starting from the state, can the model learn to attend to the right length of history? Or can we make optimal history length search as part of the training? For example, we can find the maximum length during the training and let the model only attend to states within the lengths?

About the empirical results and the contribution of the paper: EDT significantly improves the performance of DT, which is similar to the performance of IQL. Can the authors elaborate the implications of this? Should we simply use IQL since it already has the best performance? What are the reasons for using EDT? (more computationally efficient, simpler to implement?)


**Limitations:**

As the authors have mentioned in the paper, EDT adds a computational overhead in the inference process of DT by choosing the history length (Alg. 1).

---

> ### Author Rebuttal · Authors · 2023-08-08
>
> We thank the reviewer for their insightful comments. We address your comments in the following.
>
>
> ---
>
> **Q**: “Does this simply implement the idea behind the IQL algorithm in DT?”
>
> **A**: Although both IQL and EDT employ expectile regression, they use it for different purposes. In IQL, expectile regression is utilized to implicitly estimate the Q-Learning objective, specifically serving as a Q-value maximizer with respect to actions. In contrast, EDT employs expectile regression to estimate return concerning previous history, composed of observations, previous returns, and actions. This enables us to choose the optimal history length. While the loss itself may be similar, the computation and the ways we apply them are quite distinct.
>
> ---
>
> **Q**: “Need background on expectile regression.”.
>
> **A**: Thank you for your suggestions. Due to limited space, we were unable to cover more information and details regarding expectile regression. We will provide more detailed background knowledge about expectile regression in the Appendix of our revised manuscript.
>
> ---
>
> **Q**: “Minor points about clarity and missing references”.
>
> **A**: Thank you for pointing this out. We will revise the manuscript according to your suggestions.
>
> ---
>
> **Q**: "My understanding is that training always uses the same history length $T$. Does it help to use different history lengths during training as well? It seems that training and inference use histories of different lengths (always length of $T$ for training, length of $\leq T$ for inference). Would it harm the performance?"
>
> **A**: During EDT training, the $\tilde{R}$ is trained with respect to different history lengths and this can be achieved with a standard data loader and DT training. To accomplish this, we sample sub-trajectories with a context length $T$. For example, let’s say $T=3$, and the sampled input will be $[o_{t-2}, R_{t-2}, a_{t-2}, o_{t-}, R_{t-1}, a_{t-1}, o_{t}, R_{t}, a_{t}]$, with the task being to predict $[(R_{t-2}, \tilde R_{t-2}), a_{t-2}, o_{t-1}, (R_{t-1}, \tilde R_{t-1}), a_{t-1}, o_{t}, (R_{t}, \tilde R_{t}), a_{t}, o_{t+1}]$. In this case, the model is trained to predict $R_{t-2}$ and $\tilde R_{t-2}$ using only $o_{t-2}$, which corresponds to a unit history length. Due to the inherent structure of DT training, EDT naturally learns the value of $\tilde{R}$ for each potential length.
>
> ---
>
> **Q**: "Sec. 4.4 shows that different history lengths do affect the performance. I wonder if the optimal history length search can be avoided without harming the performance: Given the estimate of the maximum value starting from the state, can the model learn to attend to the right length of history? Or can we make optimal history length search as part of the training? For example, we can find the maximum length during the training and let the model only attend to states within the lengths?"
>
> **A**: Finding the optimal history length without conducting a history length search is indeed an intriguing research question. In this work, we introduce a straightforward search method to estimate the optimal length. While this approach requires slightly more inference time (a difference that is almost negligible, as shown in the table below), it effectively illustrates the idea that varying the history length enables DT to achieve trajectory stitching and derive better performance. As a potential enhancement, one could directly predict the optimal length by training a classifier on an offline RL dataset, labeled with optimal lengths as predicted by the trained EDT model. This method would allow the classifier to estimate the optimal length with an inference time roughly equivalent to that of the standard DT, simplifying the process.
>
> |               $\delta$               |    2   |    4   |    6   |    8   |   DT   |
> |:------------------------------------:|:------:|:------:|:------:|:------:|:------:|
> | wall clock time for 1000 steps (sec) | $12.6$ | $12.1$ | $11.8$ | $11.5$ | $11.4$ |
>
> ---
>
> **Q**: "About the empirical results and the contribution of the paper: EDT significantly improves the performance of DT, which is similar to the performance of IQL. Can the authors elaborate the implications of this? Should we simply use IQL since it already has the best performance? What are the reasons for using EDT? (more computationally efficient, simpler to implement?)"
>
> **A**: The experimental results show that EDT greatly improves the performance of DT, indicating that trajectory stitching is one of the major issues that hinder DT's competitiveness with other SOTA methods in single-task scenarios. Indeed, IQL is a simple yet effective Q-learning approach, but our experiments demonstrate that EDT yields superior results in multi-task learning. The Multi-Game Decision Transformers [1] also reveals that it significantly outperforms Q-learning-based approaches, such as CQL, in multi-game scenarios without directly addressing trajectory stitching. In conclusion, our experiments suggest that EDT is a more favorable option than IQL for multi-task learning.
>
> ---
>
> Please do not hesitate to let us know if you have any additional comments.
>
> [1] Lee et al., Multi-Game Decision Transformers (2022), https://arxiv.org/abs/2205.15241

---

> > ### Comment · Reviewer_ak2g · 2023-08-15
> > **Thanks for your response**
> >
> > I thank the authors for their thorough responses. My questions and concerns are addressed and I'd like to keep the score.
> >
> > The additional experiment results confirm that the search process during inference does not consume much more computation time. However, it would still be great if the Transformer can learn to determine the optimal history length (for example, learning which states to mask to generate a better trajectory). I believe this is an interesting future direction and is not necessary for this paper.
> >
> > So EDT indeed trains on trajectories of different lengths. Reading the relevant sections again, I did find one sentence talking about this in Line 164-165, “This estimation aids in comparing expected returns of different trajectories over various history lengths.” It would be great to make this clearer, possibly by describing the training algorithm using pseudocode.

---

> ### Author Response · Authors · 2023-08-15
>
> Dear Reviewer ak2g,
>
> Thank you once again for your feedback and comments on our work! We would appreciate it if you could let us know if there are any further questions or concerns.
>
> Best regards,
>
> Author 134

---

### Official Review · Reviewer_GPqH · 2023-07-10

**Soundness:** 2 fair
**Presentation:** 1 poor
**Contribution:** 2 fair
**Rating:** 4
**Confidence:** 3

**Summary:**

This paper introduces the Elastic Decision Transformer (EDT), an extension of the Decision Transformer (DT) model that automatically determines the history length input during inference. Specifically, the EDT estimates the value of each candidate history length with the expectile return regression and chooses the length that yields the maximum value. They validate the effectiveness of EDT on RL benchmarks, demonstrating that its performance surpasses that of the DTs and other offline RL algorithms, particularly in the context of multi-task scenarios.

**Strengths:**

* The concept of adaptively adjusting the history length input to the DT model is intriguing.
* The introduced solution addresses the trajectory stitching problem in the original DT model.
* The authors utilize expectile return regression to select the value of each candidate history length
* The authors demonstrate its utility in RL benchmark environments.

**Weaknesses:**

* The design and implementation of the Elastic Decision Transformer (EDT) model appear ad-hoc, with insufficient discussions on the situations where the model is effective or ineffective.
    * The selection process for history length, leaning towards an optimistic approach by possibly ignoring potentially inconvenient histories, calls into question the model's adaptability in non-Markovian or stochastic environments where the original DT might be suitable.
    * Considering the inherent attention mechanism of the Transformer architecture, it remains unclear whether the proposed method of adjusting the input history length is beneficial universally, particularly in cases of sufficient data availability.
* The presentation of the methods lacks clarity, with missing definitions for variables and loss functions. The necessity to refer to external sources ([29]) for key details detracts from the paper's self-contained nature.
    * The inclusion of an algorithm table to detail the entire process would have significantly aided reader understanding.
    * There is an apparent lack of effort to make the paper self-contained.

**Questions:**

* What history length is used during training? Is it a fixed length, or does the EDT model learn \tilde R for each potential length?
* Why isn't there a comparison with Multi-game DT[29], especially for multi-task experiments?
* Could the authors clarify line 162? In particular, what the authors mean by "we optimize the action space as the return objective L return using cross-entropy." Although I understand that the authors are estimating the return distribution with a categorical return distribution, there is no explicit mention of this, which can lead to confusion.
* The calculation process for P(R^t) in Equation (4) is unclear. Could the authors provide more details on this?
* What is "expert return distribution"? There seems to be no mention of the "expert." Moreover, the definitions of the returns R, \tilde R, and \hat R are needed.
* The definition of the loss is unclear, as it requires a reference to [29]. Also, the additional losses L_return and L_observation, which were not used in the original DT, have been added to the loss, so an explanation is necessary.
* Should "Figure 2" on line 234 be "Table 2"?

**Limitations:**

* The paper lacks a comprehensive exploration of the specific situations where the model excels or falls short. It would be valuable to see a deeper investigation into the behavior of the model across various scenarios, especially regarding the impact of history-length selection in different types of environments. In other words, if the Elastic Decision Transformer (EDT) essentially narrows down the scope of application of the original Decision Transformer (DT) by taking advantage of Markov properties, it is unclear what unique benefits it retains when compared with other comparable models like IQL or CQL.
* The history length selection process appears to be overly optimistic, possibly overlooking histories that may be inconvenient to the EDT model. This approach could limit the model's adaptability in stochastic non-Markovian environments. To address this, the authors could conduct experiments in such environments to validate the model's robustness.
* This paper isn't entirely self-contained; understanding crucial aspects of the algorithm, particularly those related to returns and losses, necessitates referencing external resources such as [29]. This undermines the paper's self-contained nature and can hinder the reader's understanding.

---

> ### Author Rebuttal · Authors · 2023-08-08
>
> We thank the reviewer for their insightful comments.
>
> ---
>
> **Q**: The design of the EDT appears ad-hoc.
>
> **A**: The components proposed in EDT such as using expectile regression to estimate the maximum return and optimal history length search, are not particularly designed for specific cases. Instead, the implementation of EDT are rather simple and straightforward since we would like to purely demonstrate the idea–namely, using a varying history length can be helpful for trajectory stitching. It's important to note that the core contribution of this work is to provide a novel method for trajectory stitching. This is a key factor in why DT-based approaches might fall behind the SOTA in single-task scenarios.
>
> ---
>
> **Q**: The selection process for history length, leaning towards an optimistic approach by possibly ignoring potentially inconvenient histories, calls into question the model's adaptability in non-Markovian or stochastic environments where the original DT might be suitable.
>
> **A**: EDT doesn't necessarily resolve the trajectory stitching issue in non-Markovian environments since we consider the Markov Decision Process in this work. We've demonstrated improvements across both locomotion and Atari tasks, in both single-task and multi-task scenarios. This work opens up the possibility for DT-based approaches to achieve comparable performance in single-task scenarios and to significantly outperform other methods in multi-game scenarios (18% improvement in locomotion tasks, 101% improvement in Atari tasks).
>
> ---
>
> **Q**: Considering the attention mechanism of the Transformer architecture, it remains unclear whether the proposed method of adjusting the input history length is beneficial universally, particularly in cases of sufficient data availability.
>
> **A**: We utilized the same number of demonstrations as those presented in D4RL, an offline RL benchmark for model evaluations. In our experiments, we demonstrated the superiority of EDT over the DT baseline in both single-task and multi-task scenarios. The improvements were significant: 27% in single-task D4RL, 44% in multi-task D4RL, and 40% in multi-task Atari . The attention mechanism of the Transformer architecture enables DT to predict actions robustly in accordance with training trajectories. However, this may be insufficient for DT to choose a better action that may differ from the decision presented in the demonstrations.
>
> ---
>
> **Q**: What history length is used for training?
>
> **A**: During the training process, the EDT model learns the values of $\tilde{R}$ for each potential length. We use the same data loader as other Decision Transformer models, and we sample sub-trajectories with a context length $T$. For example, if $T=3$, the sampled input will be $[o_{t-2}, R_{t-2}, a_{t-2}, o_{t-1}, R_{t-1}, a_{t-1}, o_{t}, R_{t}, a_{t}]$, and the output will be to predict $[(R_{t-2}, \tilde R_{t-2}), a_{t-2}, o_{t-1}, (R_{t-1}, \tilde R_{t-1}), a_{t-1}, o_{t}, (R_{t}, \tilde R_{t}), a_{t}, o_{t+1}]$. In this context, the model learns to predict both $R_{t-2}$ and $\tilde R_{t-2}$ based solely on $o_{t-2}$. Due to the inherent nature of Decision Transformer training, the EDT naturally learns the value of $\tilde R$ for each potential length.
>
> ---
>
> **Q**: Didn't compare with Multi-Game DT (MGDT).
>
> **A**: The training script for MGDT has not been released, and EDT without varying history length essentially serves as an approximation to MGDT. The main difference is that their model doesn't estimate the next state and $\tilde{R}$. Consequently, in our experiments, we compare the proposed EDT to an architecture without varying length, serving as a re-implementation of MGDT. To further illustrate this, we provide the performance of EDT without varying history length, using it as an approximation for the performance of MGDT in our multi-task experiment.
> |               | Hopper | Walker | Halfcheetah |   Ant  | Atari (Mean %) |
> |:-------------:|:------:|:------:|:-----------:|:------:|:------------:|
> | EDT (fixed T) | $56.5$ | $33.7$ |    $31.0$   | $82.5$ |    $75.5$   |
>
> ---
>
> **Q**: Explain "we optimize the action space as the return objective L return using cross-entropy."
>
> **A**: Thank you for pointing this out. We will revise our manuscript to clarify that we use tokenized values for the return.
>
> ---
>
> **Q**: The calculation process for $P(R^t)$ is unclear.
>
> **A**: To derive $P(R^t|\text{expert}^t)$, we first apply Bayes’ Rule: $P(R^t|\text{expert}^t) \propto P(\text{expert}^t|R^t)P(R^t)$. Then, we define a binary classifier that classifies whether an instance is from expert demonstrations. This classification is done with a probability proportional to the future return, using an inverse temperature $\kappa$. Similar approaches have been studied extensively in previous works [1,2,3]. As a result, we approximate $P(\text{expert}^t|R^t)$ with $\exp(\kappa R^t)$, which gives $P(R^t|\text{expert}^t) \propto \exp(\kappa R^t)P(R^t)$.
>
> ---
>
> **Q**: What is "expert return distribution"? Define the returns $R$, $\tilde R$, and $\hat R$.
>
> **A**: The term 'expert' here suggests 'optimal.' Regarding the return definition, we use $\hat R$ for normal return, $\tilde R$ for the maximum return estimator, and $R$ otherwise. We will make changes to the manuscript to enhance clarity.
>
> ---
>
> **Q**: The additional losses not used in the DT, have been added to the loss.
>
> **A**: In the paper, we mentioned that the return and observation losses are computed with cross-entropy loss and mean-square error, respectively. We also illustrate the architecture that EDT uses to predict return and the next observation in Figure 2. To improve clarity, we will modify the manuscript to highlight that we employ tokenized return and detail the loss computation.
>
> ---
>
> [1] Lee et al., Multi-Game Decision Transformers
>
> [2] Kappen et al., Optimal control as a graphical model inference problem
>
> [3] Toussaint et al., Robot trajectory optimization using approximate inference

---

> ### Author Response · Authors · 2023-08-15
>
> Dear Reviewer GPqH,
>
> Thank you once again for your feedback and comments on our work! We would appreciate it if you could let us know if there are any further questions or concerns.
>
> Best regards,
>
> Author 134

---

> > ### Comment · Reviewer_GPqH · 2023-08-20
> >
> > Thank you for your detailed rebuttal and clarification on various aspects of the EDT. Your efforts in addressing the concerns are appreciated. Regarding the selection process for history length: the method seems to lean towards an optimistic approach by using a max operation, which may not be ideal in the context of offline RL because of the issue of distributional shift. It would be beneficial to see a more in-depth discussion or investigation on how this method fares, even when the target environment is limited to MDP. Having read the other reviews and the author's responses, I am inclined to retain my initial score.

---

> > > ### Author Response · Authors · 2023-08-21
> > >
> > > Dear Reviewer GPqH,
> > >
> > > Thank you for your constructive feedback. We acknowledge that distribution shift is a critical issue in reinforcement learning, and addressing it could certainly improve the robustness of such approaches. However, the primary objective of our work is to offer an alternative and promising method for trajectory stitching. This allows offline reinforcement learning approaches, such as the Decision Transformer, to achieve significantly better performance in both single-task and multi-task scenarios. As such, addressing distribution shift might be beyond the scope of this work. Nonetheless, we recognize its importance and will consider it a promising avenue for future research.
> > >
> > > Best regards,
> > > Author 134

---

### Official Review · Reviewer_kCB8 · 2023-07-19

**Soundness:** 3 good
**Presentation:** 4 excellent
**Contribution:** 3 good
**Rating:** 6
**Confidence:** 4

**Summary:**

This paper proposes to vary the history length provided to a Decision Transformer to enable it to stitch suboptimal trajectories into better ones. The motivation behind this, the authors argue, is that the model can shift to an optimal trajectory best given only the current state (and not the history), while providing the history can better help the model maintain an optimal trajectory once found. Concretely, this method EDT trains a value function to estimate the highest achievable return given a history length, and searches for the best history length for action inference. In experiments on the D4RL benchmark and Atari games, EDT is shown to outperform DT, which they show struggles with trajectory stitching.

**Strengths:**

- The proposed method seems to work better than DT and is competitive with offline RL methods.
- The experimental results are quite thorough with multiple domains and relevant comparisons.
- EDT also seems especially strong in the multi-task setting, where Q-learning-based offline RL methods typically struggle.
- The solution is intuitive: it essentially varies the receptive field at inference time, because the model would otherwise just follow the training trajectories given the full history.

**Weaknesses:**

- The main limitation of the method seems to be inference cost, as it requires a search over all possible history lengths at each time-step. While the search space can be reduced by defining a coarser range of possible history lengths, this can also compromise the performance of the method.
- It would be helpful to study a separate setting which _only_ requires stitching of offline trajectories (maybe a gridworld example?) to see how EDT compares to Q-learning-based offline RL more directly.
- In the introduction, the messaging could be a bit clearer by mentioning that this observation applies to DT / sequence modeling approaches.

**Questions:**

- Are the plots in Fig. 7 illustrating the best history length according to the value estimator?
- If so, I’m curious what the performance of DT would look like if we set the history length to the mode of the distributions shown in Fig. 7 (i.e., either 3 or 4). Are the results in Table 3 illustrating this but for history length 5 and 20? Or, are there other differences between DT and EDT?
- How much data is needed to train the value estimator? What happens if you train the estimator on a subset of the DT data or on a superset of the DT data? This would help get a sense for whether the data requirement of the estimator is greater than or smaller than the DT.
- What does "-ONE" indicate?

Nits:
- Could error bars be provided for the Atari results?
- Line 212: casual -> causal


================= POST-REBUTTAL RESPONSE ===================

Thank you for the detailed response. I still am in favor of acceptance.

**Limitations:**

Yes

---

> ### Author Rebuttal · Authors · 2023-08-08
>
> We thank the reviewer for their insightful comments. We address your comments in the following.
>
> ---
>
> **Q**: “The main limitation of the method seems to be inference cost”
>
> **A**: Thank you for pointing this out. We have conducted an ablation study to compare the wall clock time with the proposed $\delta$ during the history length search. The following results suggest that although EDT indeed requires additional time for searching the length, the difference is quite small when compared with sampling from environments like gym. The searching approach can be potentially improved by learning a predictor to accelerate the inference speed.
>
> |               $\delta$               |    2   |    4   |    6   |    8   |   DT   |
> |:------------------------------------:|:------:|:------:|:------:|:------:|:------:|
> | wall clock time for 1000 steps (sec) | $12.6$ | $12.1$ | $11.8$ | $11.5$ | $11.4$ |
>
> ---
>
> **Q**: “It would be helpful to study a separate setting which only requires stitching of offline trajectories (maybe a gridworld example?) to see how EDT compares to Q-learning-based offline RL more directly.”
>
> **A**: Directly visualizing the trajectory results using a simpler task, such as gridworld, and comparing the results with a Q-learning-based approach is indeed an interesting idea. In a simple gridworld scenario, the Q-learning-based method might choose to go for the grids with the highest estimated return. On the other hand, vanilla DT may directly follow the paths presented in demonstrations. EDT, however, might show a pattern similar to the Q-learning-based approach. An extreme case of this is when EDT adopts a unit history length throughout action inference. This can result in identical results to the Q-learning-based approach, assuming the learned Q-values and returns are closely matched.
> However, due to limited time, we were unable to provide a comprehensive visualization at this stage. We plan to work on visualizing the learned policy in this direction for a future version and will include these insights on our project website in the future.
>
> ---
>
>
> **Q**: “In the introduction, the messaging could be a bit clearer by mentioning that this observation applies to DT / sequence modeling approaches.”
>
> **A**: Thank you for your suggestions. We will improve the clarity of our manuscript in the introduction section.
>
> ---
>
>
> **Q**: Are the plots in Fig. 7 illustrating the best history length according to the value estimator?
>
> **A**: Yes. In Figure 7, we plotted the distribution of the best history length in trajectories.
>
> ---
>
> **Q**: If so, I’m curious what the performance of DT would look like if we set the history length to the mode of the distributions shown in Fig. 7 (i.e., either 3 or 4). Are the results in Table 3 illustrating this but for history length 5 and 20? Or, are there other differences between DT and EDT?
>
> **A**: Table 3 illustrates EDT's performance when the history length is fixed to 5 and 20. Since we observed that some datasets prefer a shorter 'fixed history length' while others favor a longer one, we provide a thorough comparison by presenting the performance for history lengths of 5 and 20. Additionally, in the table below, we show EDT's performance when the history length is set to 3. These results suggest that although a shorter history length may be preferred based on the outcome of history length search, it can also be critical to use a longer history length when necessary. This demonstrates EDT's ability to utilize history length appropriately.
>
> |                   |    hopper-m   |    hopper-mr   |    walker-m   |    walker-mr   | halfcheetah-m | halfcheetah-mr |
> |:-----------------:|:-------------:|:--------------:|:-------------:|:--------------:|:-------------:|:--------------:|
> | EDT (fixed $T=3$) | $60.5\pm 4.6$ | $57.8\pm 22.7$ | $68.3\pm 8.1$ | $57.0\pm 14.6$ | $42.1\pm 0.4$ |  $16.6\pm 4.2$ |
>
> In the table, ‘m’ suggests ‘medium’ and ‘mr’ suggests ‘medium-replay’.
>
> The differences between DT and EDT can be summarized as follows:
> During training, we additionally optimize the expectile regression loss in EDT. This allows us to estimate the maximum return a history length can possibly achieve.
> During action inference, EDT utilizes the estimated values from the approximate return maximizer, obtained through expectile regression, to identify the optimal history length.
>
>
> ---
>
> **Q**: How much data is needed to train the value estimator? What happens if you train the estimator on a subset of the DT data or on a superset of the DT data? This would help get a sense for whether the data requirement of the estimator is greater than or smaller than the DT.
>
> **A**: We conducted an ablation study on the size of demonstrations, where the ratio indicates the proportion of demonstrations used for training. For this experiment, we used the 'hopper-medium-replay' demonstration, and the results are summarized in the table below. The results show that EDT still outperforms the DT baseline with a sufficient amount of demonstrations. However, when the demonstration ratio falls below 0.4, both methods fail to learn meaningful policies. An interesting observation is that although EDT outperforms the baseline DT when the demonstration ratio is greater than or equal to 0.6, the difference becomes less significant when the ratio is less than 0.8. This indicates that having enough demonstrations is one of the key requirements to benefit from trajectory stitching in EDT.
>
> | ratio |   1.0  |   0.8  |   0.6  |  0.4  |  0.2  |
> |:-----:|:------:|:------:|:------:|:-----:|:-----:|
> |  EDT  | $89.0$ | $65.2$ | $19.6$ | $5.3$ | $5.4$ |
> |   DT  | $61.9$ | $59.7$ | $18.2$ | $5.1$ | $5.5$ |
>
> ---
>
> **Q**: What does "-ONE" indicate?
>
> **A**: The “-ONE” denotes that we employ a single model to learn all tasks in a multi-task scenario, distinguishing it from the setting used in the single-task experiments.
>
> ---
>
> Please do not hesitate to let us know if you have any additional comments.

---

> > ### Comment · Reviewer_kCB8 · 2023-08-14
> > **Response to authors**
> >
> > Thank you for the detailed response. I still am in favor of acceptance.

---

### Author Rebuttal · Authors · 2023-08-08

We thank all reviewers for their thoughtful comments and insights. We will revise our manuscript based on your feedback, and we have responded to each of your individual comments.

Summary of revisions:
 * Fixing typo: “casual“ -> “causal“, “Figure 2“ -> “Table 2“ on line 234.
 * Adding explanation regarding tokenized return and details about return and observation loss computation.
 * Including notes about the interchangeable use of terms ‘expert’ and ‘optimal’ and the definition of returns ($R$, $\hat R$, $\tilde R$).
 * Adding related work Prompt DT and Hyper DT.
 * Implementation details and background knowledge regarding the expectile regression.
 * Improving clarity of the introduction section.

We would also like to provide a comparison of the training and inference times in terms of wall clock time. This demonstrates that the additional computational cost induced by the expectile regression loss and the optimal history length search is not significant. The experiments were conducted on a 3090 GPU, and the results show that the training time remains almost unchanged. Meanwhile, the additional action inference time constitutes only around 10% of the total inference time since the sampling from the gym environment takes considerably more time than action inference.

Wall clock time comparison (training):
|                                            |   DT   |   EDT  |
|:------------------------------------------:|:------:|:------:|
| wall clock training time (500 epochs, sec) | $1442$ | $1449$ |

Wall clock time vs. $\delta$ (action inference):
|               $\delta$               |    2   |    4   |    6   |    8   |   DT   |
|:------------------------------------:|:------:|:------:|:------:|:------:|:------:|
| wall clock time for 1000 steps (sec) | $12.6$ | $12.1$ | $11.8$ | $11.5$ | $11.4$ |

Again, we thank the reviewers for their constructive feedback. We believe that all comments have been addressed, but are happy to address any further comments from reviewers.

Best,
Author of 134

---

### Decision · Program_Chairs · 2023-09-21

**Decision:**

Accept (poster)

**Comment:**

Decision transformers are a recent development in offline RL that aim to turn offline RL into a sequence modeling problem. In principle, DTs can "stich" the optimal parts of suboptimal trajectories together to create optimal behavior. In practice however, trajectory stiching is not found to empirically take place. This paper proposes a new method for training and deploying DTs (called Elastic Decision Transformer) and shows that it is better suited for stiching suboptimal trajectories together. Thus the proposed approach is a step towards realizing the potential benefits of DTs.

Main strengths: The proposed method is an intuitive modification to decision transformers that improves empirical performance and the paper includes a comphrensive empirical evaluation.

Main Concerns: The proposed method increases the latency of action selection using the trained transformer. The author rebuttal included wall clock time measurements that show the increase is significant but. not very large.

Decision transformers have recently become popular as a means to bring the power of sequence modeling to offline RL. This paper takes a step towards realizing some of the potential of DTs and thus I think it makes a useful contribution to the RL research community.